# Application of a Novel CVD TiN Coating on a Biomedical Co–Cr Alloy: An Evaluation of Coating Layer and Substrate Characteristics

**DOI:** 10.3390/ma13051145

**Published:** 2020-03-05

**Authors:** Si Hoon Song, Bong Ki Min, Min-Ho Hong, Tae-Yub Kwon

**Affiliations:** 1Department of Medical & Biological Engineering, Graduate School, Kyungpook National University, Daegu 41940, Korea; song@taegutec.co.kr; 2Center for Research Facilities, Yeungnam University, Gyeongsan 38541, Korea; 3Department of Dental Laboratory Science, College of Health Sciences, Catholic University of Pusan, Busan 46252, Korea; mhhong@cup.ac.kr; 4Department of Dental Biomaterials, School of Dentistry and Institute for Biomaterials Research & Development, Kyungpook National University, Daegu 41940, Korea

**Keywords:** cobalt–chromium alloy, titanium nitride, physical vapor deposition, chemical vapor deposition, microstructure

## Abstract

Titanium nitride (TiN) was deposited on the surface of a cobalt–chromium (Co–Cr) alloy by a hot-wall type chemical vapor deposition (CVD) reactor at 850 °C, and the coating characteristics were compared with those of a physical vapor deposition (PVD) TiN coating deposited on the same alloy at 450 °C. Neither coating showed any reactions at the interface. The face-centered cubic (fcc) structure of the alloy was changed into a hexagonal close-packed (hcp) phase, and recrystallization occurred over at 10 μm of depth from the surface after CVD coating. Characteristic precipitates were also generated incrementally depending on the depth, unlike the precipitates in the matrix of the as-cast alloy. On the other hand, the microstructure and phase of the PVD-coated alloy did not change. Depth-dependent nano-hardness measurements showed a greater increase in hardness in the recrystallization zone of the CVD-coated alloy than in the bulk center of the alloy. The CVD coating showed superior adhesion to the PVD coating in the progressive scratch test. The as-cast, PVD-coated, and CVD-coated alloys all showed negative cytotoxicity. Within the limitations of this study, CVD TiN coating to biomedical Co–Cr alloy may be considered a promising alternative to PVD technique.

## 1. Introduction

Among various biomaterials, cobalt–chromium (Co–Cr) alloys have been widely used as commercial orthopedic implant materials due to their excellent biochemical properties, moderate strength, and their resistance to corrosion and abrasion. Among these properties, the biochemical stability of the alloys is known to be due to the passive film on their surface [1,2]. Due to long-term fatigue friction and the subsequent wear of orthopedic implants, however, this passive film may become unstable and, as a result, allow metal ions released from the alloys into the body fluids (serum, urine, etc.) to accumulate between the implant and the tissues, causing an inflammatory reaction in the surrounding tissue [3,4]. Such metal ions and wear debris, concentrated at the implant–tissue interface, may migrate through the tissue, resulting in clinical implant failure, osteolysis, cutaneous allergic reactions, and remote site accumulations [4].

Given the increased clinical use of Co–Cr alloys as an implant material, many attempts, including ceramic coating, have been made to compensate for these problems. Among these, various physical vapor deposition (PVD) titanium nitride (TiN) coating methods, such as sputtering and arc-ion plating, have been extensively studied and are now commercially available [5,6]. A few microns of TiN coating increases the surface hardness of the implant, improves the mechanical durability and fatigue resistance by reducing the friction coefficient, and effectively prevents metal ions from leaching into the surrounding tissue by creating a dense film resistant to acids and bases [7,8,9,10,11,12]. Since PVD is a directional deposition method, it may not uniformly cover all surfaces of the implant due to the shadow effect and often creates variations in coating thickness due to poor step coverage. In addition, the method cannot effectively coat the entire inner surface of the implants. The coated layer formed by PVD at low temperature has fundamentally limited adhesion. In particular, the strong stress inevitably generated during the coating procedure may cause peeling and buckling of the coating layer at the sharp edges of the implant [13].

Chemical vapor deposition (CVD) can compensate for the above main disadvantages of PVD technique. CVD can cover all surfaces (inner surfaces as well as outer) of the implant with a relatively uniform thickness. If the thermal expansion coefficient of the implant is greater than that of the coating material, CVD can generate a strong compressive stress on the surface of the implant, enhancing the adhesion. The high purity gaseous raw materials allow high purity coatings to be made. In addition, the coating morphology and specific surface area can be controlled by deposition conditions [14,15]. However, the high coating temperature and some corrosive gas by-products generated during the coating process may cause an undesirable reaction at the implant–coating interface depending on the chemical composition of the implant substrate and its reactivity. Recently, von Fieandt et al. [16] studied the phase content, growth rate, morphology, and microstructure of the CVD of TiN on three different pure transition metal substrates (Fe, Co, and Ni) from a TiCl_4_-N_2_-H_2_ gas system. To date, however, little research has been done on the application of CVD TiN coating to the Co–Cr alloy used as an orthopedic implant material.

In this study, TiN was coated on a Co–Cr-based alloy using hot-wall type CVD equipment and the quality of the coating layer as well as potential factors affecting the physical properties of the coating-substrate interface were investigated. CVD coating characteristics were also compared with those of a commercially available PVD coating.

## 2. Materials and Methods

### 2.1. Specimen Preparation

Disc-shaped Co–Cr alloy specimens (10 mm in diameter and 3 mm in thickness) were prepared using casting. A disc model was designed with computer-aided design (CAD) software (AutoDesk Inventor 2015, Student Version, Autodesk, San Rafael, CA, USA), and wax patterns (VisiJet^®^ M3 Dentcast, 3D Systems, Rock Hill, SC, USA) were prepared from the CAD data using a multi-material 3D Printer (ProJet^®^ 5500X, 3D Systems, Rock Hill, SC, USA) [17]. The patterns were embedded in a phosphate-bonded investment material (Univest Non-Precious, Shofu Inc., Kyoto, Japan), and were then cast using the Co–Cr alloy (StarLoy C, DeguDent, Hanau-Wolfgang, Germany; composition (in wt%): Co 59.4, Cr 24.5, W 10, Nb 2, V 2, Mo 1, Si 1, Fe 0.1) with a centrifugal casting apparatus (Centrifico Casting Machine, Kerr Corp., Orange, CA, USA) [17]. One flat surface of each specimen was mirror-polished with wet silicon carbide paper (up to 2000-grit), with 9, 3, and 1 μm diamond suspensions (Allied High Tech Products, Rancho Dominguez, CA, USA), and finally with 50 nm colloidal silica (Allied High Tech Products) on a polishing cloth using a rotary grinding/polishing machine (M-Prep 3, Allied High Tech Products). All of the polished specimens were cleaned with ethanol in an ultrasonic water bath for 20 min prior to coating.

### 2.2. PVD and CVD Coating

The disc Co–Cr alloy specimens were subjected to either PVD- or CVD-coating processes (Table 1). The PVD coating was performed in a commercial arc-ion plating (AIP) reactor. The TiN coating was deposited at 550 °C. After the deposition temperature was reached in an argon atmosphere, the surface was etched to enhance coating adhesion. During deposition of TiN, a 30-volt bias was applied to the specimens using a Ti target and N_2_ gas for 30 min.

The CVD TiN coating was performed using a hot-wall type CVD reactor in which all gas switchgear and flow rates were controlled by a computer system. Figure 1 shows a schematic diagram of the CVD reactor, in which N_2_ and H_2_ were fed into the gas mixing chamber in a gaseous state from reservoirs (a) and (b), respectively. TiCl_4_ was passed from the reservoir (c) to a liquid state via a liquid mass flow meter, fed into a mixer, vaporized inside the mixing chamber, and finally mixed with other gases. The mixed gas fed into the furnace was uniformly applied in the holes of a rotating graphite tube (h) located in a straight line from the bottom to the top of the coating chamber. The unreacted gases and byproducts were exhausted and collected via a liquefying apparatus (i) and vacuum pump (j). All the inside of the reactor including the graphite fixtures had been pre-coated with TiN to eliminate the influence of carbon during the deposition process. The TiN coating was deposited at 850 °C to minimize thermal deformation and change in the physical properties of the alloy while obtaining a proper coating thickness. The furnace was heated from room temperature to the deposition temperature of 850 °C in a hydrogen atmosphere. After that, TiN was deposited for 400 min at a specified gas ratio. The furnace was then fast-cooled down to 300 °C using a hydrogen gas and allowed to slow-cool to room temperature. Excess hydrogen was used as a carrier for transporting other gases and, at the same time, to sequentially dissociate Cl from TiCl_4_, allowing Ti ions to react with N_2_. In addition, the amount of N_2_ was determined by considering the reaction rate and activation energy with TiCl_4_ [18]. The main reaction scheme of TiN coating in the reactor during deposition is as follows [16]: TiCl_4_ (g) + 1/2N_2_ (g) + 2H_2_ (g) → TiN (s) + 4HCl (g).

### 2.3. Characterization

For the Co–Cr alloy substrate and either PVD or CVD TiN-coated specimens, phase identification was carried out by X-ray diffractometry (XRD, X’Pert PRO, PANalytical, The Netherlands), using Cu K_*α*_ radiation (*λ* = 0.1541 nm) at an accelerating voltage of 40 kV, a beam current of 30 mA, a 2θ-angle scan range of 20° to 130°, a scanning speed of 2°/min, a sampling pitch of 0.02°, and a preset time of 0.6 s [19].

The disc specimens before and after coating were cross-sectioned, and the sectioned surfaces were observed using field emission scanning electron microscopy (FE-SEM, Merlin, Carl Zeiss AG, Germany) after mirror polishing as described above. To analyze the elemental composition and gains associated with the microstructure for the region close to the surface of the alloy, the specimens for each group were examined using FE-SEM with energy-dispersive X-ray spectroscopy (EDS, X-MAX, Oxford Instruments, Abingdon, UK) under an accelerating voltage of 20 kV. To determine the crystal phase and crystallographic orientation, electron backscattered diffraction (EBSD) scans were performed on the FE-SEM equipped with a Nordlys Nano EBSD detector (Oxford Instruments). The EBSD data were analyzed using AZtec (Version 3.4, Oxford Instruments) and Channel 5 (Version 5.12.72.0, Oxford Instruments) software.

To determine potential change in the mechanical properties of the Co–Cr alloy substrate after coating, the nano-hardness of the region close to the surface in the cross-sections was measured using a table top nano-indentation testing machine (CHQ11-NV10, Nanovea, Irvine, CA, USA). Nano-indentations with an area of 60 × 60 μm were made at 10, 25, and 45 μm from the surface and also the bulk center of each specimen. The maximum load was 50 mN, the indentation rate was 0.67 mN/s, and the indentation speed was 33 nm/s. The results were statistically analyzed using two-way analysis of variance (ANOVA), followed by Tukey’s post hoc test at *α* = 0.05.

To evaluate the adhesion between the coating and alloy, a progressive scratch test was performed on the coated alloy surface using a diamond cone (Revetest Scratch Tester, CSM Instruments, Needham, MA, USA). The load was progressively raised from 1 to 30 N with a range of 2 mm (progressive test mode) and a speed of 6 mm/min.

### 2.4. In Vitro Cytotoxicity Test

The cytotoxicity of the alloys was examined using the agar-diffusion method. A positive control of rubber latex and a negative control of high-density polyethylene (HDPE) with the same dimensions as the alloy specimen were also assayed. A total of 10 mL of a 4 × 10^4^ L929 fibroblast cell suspension were seeded in 100-mm dishes and incubated at 37 °C and 5% CO_2_ for 48 h. The medium was replaced with 10 mL freshly prepared agar/nutrient medium. A total of 10 mL of neutral-red (Sigma, Madrid, Spain) solution was placed on the agar surface for 10 min in the dark. After the excess dye was removed, the alloy specimens, together with the positive and negative controls, were placed on the agar surface and the dishes were incubated for 24 h. Cytotoxicity was determined by measuring the zone of cell inhibition around the specimens. In addition, the cell cultures around the specimens were then examined under a phase-contrast microscope.

## 3. Results and Discussion

### 3.1. Phase Identification

Figure 2 shows the XRD patterns of the Co–Cr alloy used as the substrate for PVD or CVD coating and TiN-deposited alloys. The matrix phase of the cast Co–Cr alloy is known to be a mainly Co- and Cr-rich *γ-* (face-centered cubic, fcc) phase and that the precipitate is an *ε* (hexagonal close-packed, hcp) phase containing W and Nb in the Co–Cr matrix. In this study, typical *γ*- and *ε*-phase diffraction patterns were detected in the Co–Cr alloy substrate (Figure 2a). All of the diffraction peaks were indexed by the diffraction peaks of the *γ*-phase (ICSD: PAN 98-007-2476) and the *ε*-phase (ICSD: PAN 98-008-7135), additionally taking into account the peak displacements and the indexing data of previous XRD studies [17,20,21]. In addition, the peak appearing near 37.7° was indexed by the Nb-rich *γ*-phase (ICSD: PAN 98-008-6919) [17]. The peaks that arbitrarily present without being affected by other peaks at the *γ*-phase were (111), (200), and (222), the peak with the highest diffraction intensity being (111). The peaks that appeared on the precipitated *ε*-phase were (010), (011), and (013), the peak with the highest intensity being (011).

In the PVD specimen (Figure 2b), the intensity of the TiN coating (200) was significantly higher than that of the matrix, and the intensity ratio of the coating/substrate was much higher than that of the CVD TiN-coated alloy (Figure 2c). Such phenomena may have been due to the highly textured one-directional coating and the relatively smooth coating surface.

In the CVD TiN-coated specimen (Figure 2c), the intensity of the peaks, including the matrix *γ*-phase (111) peaks (positioned on the three vertical dotted lines), was significantly reduced, with only the *ε*-phase and TiN (ICDD: 03-065) being detected. The X-ray diffraction has a limited penetration depth when diffracted into a material and, in most cases, presents only the surface properties of the material. The intensity depends on the state of the surface, the density of the material, crystallinity, and the absorption rate indicated by the wavelength of the X-ray. The diffraction pattern of the CVD TiN-coated specimen indicates that at least the phases below the TiN-coating layer were mainly composed of the *ε*-phase, suggesting that *γ* → *ε* phase transformation or the precipitation of the *ε*-phase occurred during the CVD coating.

### 3.2. Microstructure

Figure 3 shows the SEM images of the as-cast Co–Cr alloy and its EDS mapping images. The mirror-polished surface of the alloy showed a dendritic structure, composed of the matrix (dark) and precipitates (bright), with some small pores (black) [17,22]. The EDS mapping images showed the matrix of the alloy to be mainly composed of Co–Cr. Most of the secondary phase precipitates with a dendritic structure consisted of W-Nb-rich Co–Cr. As indicated by the arrows in (Figure 3d–f), Nb and V-based precipitates without W were observed as another secondary phase.

Figure 4 shows the low- and high-magnification SEM images of TiN deposited on the mirror-polished Co–Cr alloy surfaces using PVD or CVD. In the case of PVD TiN, very small grains (less than 100 nm) were deposited on the Co–Cr alloy surface. On the other hand, the CVD coating formed a rougher surface than the PVD coating. Lenticular-shaped grains with an average length of 200 nm and a width of 50 nm grew on the CVD-coated alloy surface.

The cross-sectional SEM images (Figure 5) show the difference between the PVD and CVD-coating layers formed on the mirror-polished Co–Cr alloy substrate. The CVD coating formed a thicker (1.5 µm) layer than the PVD coating (0.9 µm) on the substrate. In the case of PVD coating methods such as sputtering or arc-ion deposition, surfaces with an angle different from the surface facing the coating target may manifest a difference in coating thickness due to the straightness of the deposition direction of the coating material. The PVD coating layer showed very small columnar grains and irregular-shaped grains, with a clean interface between the coating and alloy devoid of chemical reaction. In the CVD coating, the growth of columnar-shaped grains created a very dense coating layer on the surface, with only a few nano-sized pores and very small secondary particles at the coating/substrate interface. One of the main differences between the two types of coatings was the microstructure near the surface of the alloy. In the PVD coating, the microstructure of the alloy near the surface resembled that of the as-cast (uncoated) alloy. On the other hand, a large amount of small pit-shaped precipitates was found in the alloy grains underneath the CVD TiN coating layer.

Figure 6 shows the cross-sectional SEM images of the CVD TiN-coated Co–Cr alloy substrate observed to a depth of 75 µm from the coating interface. In the CVD-coated alloy, unlike the as-cast and PVD-coated alloy, the SEM showed three different types (regions) of the characteristic microstructure near the interface between the coating and the alloy depending on the depth from the interface: (1) a region 12 μm deep showing small grains and pit-shaped irregular precipitates inside them (Figure 6b); (2) a region 16 to 36 μm deep having a small amount of pit- and string-shaped precipitates formed in a lattice pattern underneath the Figure 6b region (Figure 6c); and (3) a region 53 to 73 μm deep showing a small amount of pit-shaped precipitates and a scratch-like long pattern extending in one direction (Figure 6d). In particular, small grains formed in the region nearest the interface (Figure 6b).

Figure 7 shows the cross-sectional SEM images (up to 20 μm from the interface) of the three groups (as-cast, PVD-coated, and CVD-coated alloy) and their corresponding EDS mapping images. An EDS point analysis was carried out to determine the composition of the characteristic precipitates. The microstructure and precipitate distribution of the as-cast Co–Cr alloy and PVD-coated alloy were similar. Near the surface of the CVD TiN-coated alloy, there was a large distribution of Cr along the grain boundaries of the *ε*-phase and also along the grain boundaries between the dendrite and matrix. The white pits and lattice-patterned string-shaped precipitates below the pits near the coating surface (see also Figure 6b,c) had a higher W content than the bulk matrix portion. In addition, black pits, whose color was similar to or darker than the matrix, were distributed around the white pits.

Table 2 shows the results of the EDS point analysis for the points marked in images (a), (e), and (i) of Figure 7. The substrate phase of the as-cast and PVD TiN-coated alloys showed similar compositions in similar microstructure regions, the scratch-like or white lines showing the same composition as in the matrix phase (points 3 of (a) and (e)). In the CVD-coated alloy (i), the composition of point 3, showing a matrix of small grains approximately 10 μm below the interface, was not significantly different from that of point 1. Although the as-cast alloy also showed some Cr-rich area between the dendrite and matrix as well as in the matrix (see also Figure 3c), the frequency of Cr-rich area was sparse relative to the CVD TiN-coated alloy. A compositional analysis of points 6 and 7 of (i) revealed little difference in the amounts of matrix Co and Cr, while the amount of W and V was relatively high in the white pits and in the black pits, respectively. In this point analysis of the pits, the analysis range of the EDS was larger than the size of the pits, so that the results also reflect the surrounding composition. In fact, the white pits and the black pits looked like intermediate phases of being transformed into W-Nb and Nb-V precipitates, respectively. This is probably because the CVD coating was performed in the aging temperature range where precipitation or phase transformation occurs in Co–Cr alloys (Table 1).

### 3.3. Crystallographic Features

Figure 8 shows the cross-sectional EBSD images (band contrast (BC) and the phase and inverse pole figure (IPF) maps) of the as-cast, PVD-coated, and CVD-coated alloys, including the coating layer and the alloy interiors. In the BC maps, the as-cast and PVD-coated alloy specimens showed no grain boundaries other than those of the matrix and precipitates. In the CVD-coated alloy, however, very small grains were observed near the surface, together with the grain boundaries of the matrix and the precipitate. In the phase maps of the as-cast and PVD-coated alloy specimens, the matrix (blue) was the *γ*- (fcc) phase, and the precipitate (red) was the *ε*- (hcp) phase. In images (Figure 8d–f), the PVD-coated alloy showed the same microstructure as the as-cast alloy. In the CVD-coated alloy, however, the phase structure of the precipitate was the same as that of the as-cast alloy, whereas the *ε*-phase region was continuously formed from the coating surface to a depth of approximately 10 to 20 μm. In addition, scratch-like *ε*-phase patterns, not extending to the inside of the precipitated W-Nb-rich *ε*-phase, were observed in the *γ*-phase region. The length and frequency of the *ε*-phase patterns suggests a plate-like martensitic transformation [23,24]. Moreover, another Cr-rich *ε*-phase (yellow), randomly scattered inside the *ε*-phase, consisted of small grains located near the surface. In the IPF maps of all specimens, the main matrix of the alloy was all single grains within the range of the image (117 × 87 μm). In addition, the IPF maps of all alloys indicate that the W-Mo-Nb-rich precipitates shared similar orientations over a large area, suggesting that the precipitation occurred according to the crystal orientation properties of the matrix *γ*-phase when the cast alloy was formed. In Figure 8i, the *ε*-phase grains concentrated near the coating surface were randomly distributed with no specific orientation, the grain size gradually increasing from the vicinity of the coating surface to the depth direction.

The relationship between these phases and microstructures can be seen in more detail in Figure 9. In the IPF map of the CVD-coated alloy, the coated TiN maintained homogeneous crystal orientation as it grew into columnar grains. The phase map clearly confirms why the diffraction pattern of the CVD-coated alloy showed mainly the *ε*-phase (Figure 2), which seems to have originated at the interface between the *γ*-phase matrix and the TiN coating material during CVD coating. The overall microstructural development of the *ε*-phase observed near the coating surface may be interpreted as the result of phase transformation and recrystallization which simultaneously proceeded in the depth direction from the surface to the body of the alloy. These results are different from the general pearlite- or plate-like martensitic phase transformation ((111) *γ*//(0001) *ε*) of Co–Cr alloys when the aging treatment was proceeded [25]. Previous EBSD studies also showed a continuously developed *ε*-phase near the porcelain-fused-to-metal (Co–Cr alloys) interface [22,26,27]. The Cr-based *ε*-phase (yellow) in the phase map were indexed as the long *ε*-phase of the a and c axes (a: 3.7441; c: 12.1493) rather than the crystal lattice values (a: 2.5060; c: 4.0690) of W-Mo-Nb-rich precipitate dendrites. The grains in the phase map exactly overlapped with the EDS mapping image of Cr in Figure 9b. Most of the Cr-based *ε*-phases at the grain boundaries indicate that the diffusion-based phase transformation during the CVD.

### 3.4. Vickers Hardness

Figure 10 shows the Vickers hardness (Hv) values converted from the nano-indentation hardness values. The size of the indentations at 50 mN was about 3.5 to 4.5 μm in length, enabling hardness measurement of the phase transformation region of the CVD TiN-coated alloy. All indentations were made outside the W-Mo-Nb-rich *ε*-phase precipitates, with a parallel interval of about 10 μm. The hardness values of the CVD-coated alloy were higher than those of the as-cast and PVD-TiN-coated alloy at the three depths (*p* < 0.05), except at the bulk center of the specimen (*p* > 0.05). In particular, at a depth of 10 μm, the average Hv value of the CVD TiN-coated alloy (Hv 972) was much higher than that of the as-cast alloy (Hv 618), probably because all the *γ*-phases were transformed into *ε*-phases with high hardness and the grain size was reduced by recrystallization. In addition, the CVD TiN-coated alloy showed a significantly higher hardness value than those of the as-cast and PVD-coated alloys, even at a depth of 45 μm. This is probably due to the development of the plate-shaped *ε*-phase by the local martensitic transformation during the coating process and the small intermediate phase grains precipitated in the form of pits. The hardness (Hv 588) of the W-Mo-Nb-rich *ε*-phase precipitates in the as-cast alloy (points 2 of Figure 7a,e,i) was much higher than that (Hv 478) of the *γ*-phase.

### 3.5. Adhesion of the Coating

Figure 11 shows the results of the progressive scratch test (load ranged from 1 to 30 N) performed to evaluate the adhesion of the coating on the alloy. On the PVD-coated alloy (Figure 11a), surface cracks started from a 10 N load, and a progressive increment of the load increased the crack formation. The peeling and breakage of the coating grew more severe near the scratch mark as the load was progressively increased. On the CVD coating (Figure 11b), in contrast, no cracking and peeling phenomenon was found and only plastic deformation of the coated substrate was noted. The results show that the adhesion of the CVD coating was much superior to the PVD coating, even though the CVD produced a rougher surface than the PVD (Figure 5). The high temperature coating conditions and chemical reactions at the coating and alloy interfaces (Figure 1), as well as the gradient in hardness near the interface (Figure 10), may have enhanced interfacial adhesion of the CVD coating.

### 3.6. In Vitro Cytotoxicity

In the agar-diffusion test, no cell inhibition zones were seen around the negative control (HDPE) and the three alloys specimens. A clear inhibition zone was found only around the positive control (rubber latex). Moreover, cell cultures around the negative control (Figure 12b) and alloy specimens showed neutral red-stained living cells (Figure 12c–e). These findings suggest that neither the negative control nor any test specimens (including the CVD TiN-coated alloy) were cytotoxic.

## 4. Conclusions

In this study, for the first time to our knowledge, TiN was coated to a commercial biomedical Co–Cr alloy by a hot-wall type CVD apparatus. The characteristics of the coating layer, interfacial layer, and substrate were compared with those of as-cast and PVD TiN-coated alloys. The deposition of TiN on the Co–Cr alloy by CVD created a uniform coating layer, developed phase-transformed and recrystallized small grains near the alloy surface, and increased the hardness of the area. On the other hand, the PVD-coated alloy did not show any microstructural or phase changes. The findings of this study suggest that CVD TiN coating to biomedical Co–Cr alloy may be considered a promising alternative to PVD techniques, though it should be further optimized in order to avoid corrosion or unwanted phases. Further research on CVD TiN coating to biomedical alloys is needed to confirm the cytocompatibility and efficacy of the novel coating method.

## Figures and Tables

**Figure 1 materials-13-01145-f001:**
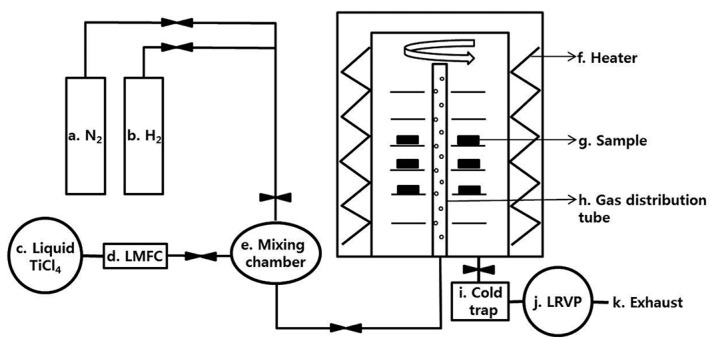
Schematic illustration of the CVD furnace used: a: N_2_ feeder; b: H_2_ feeder; c: liquid TiCl_4_ feeder; d: liquid mass flow controller (LMFC); e: TiCl_4_ vaporization and gas mixing apparatus; f: heating element; g: specimen; h: rotary type gas distribution tube; i: gas by-product liquefier; j: liquid rotary vacuum pump (LRVP); and k: exhaust of the neutralized material.

**Figure 2 materials-13-01145-f002:**
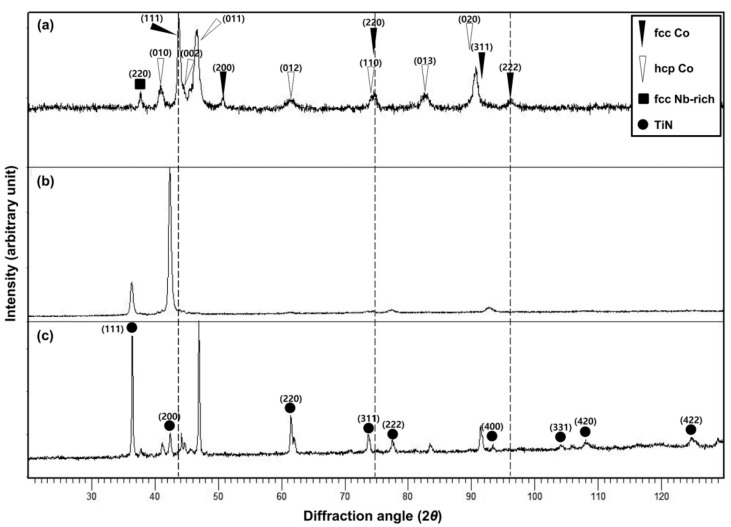
X-ray diffractometry (XRD) patterns and phases of the mirror-polished Co–Cr alloy (substrate) specimen (**a**) and the PVD (**b**) and CVD (**c**) TiN-deposited specimens.

**Figure 3 materials-13-01145-f003:**
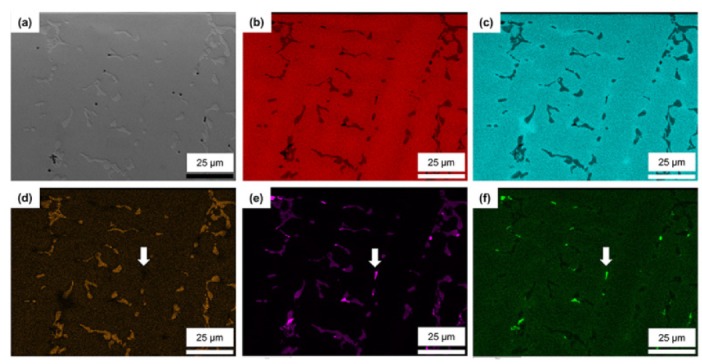
Scanning electron microscopy (SEM) micrographs of the mirror-polished as-cast Co–Cr alloy (**a**) and its energy-dispersive X-ray spectroscopy (EDS) mapping images (**b**–**f**): (**b**) Co; (**c**) Cr; (**d**) W; (**e**) Nb; and (**f**) V. Nb and V-based precipitates without W (arrow in (**d**)) were observed as another secondary phase (arrows in (**e**) and (**f**)).

**Figure 4 materials-13-01145-f004:**
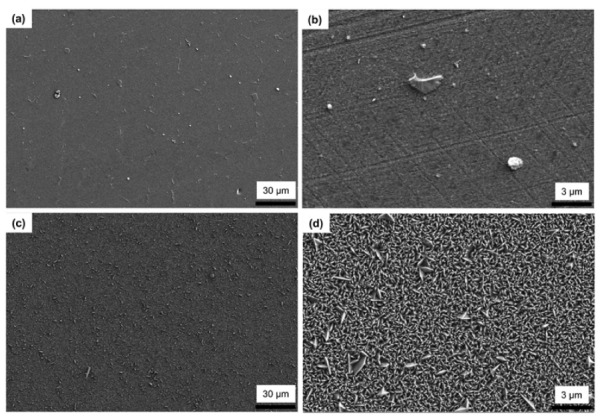
SEM micrographs showing TiN deposited on the mirror-polished Co–Cr alloy surfaces by PVD and CVD. (**a**) and (**b**): PVD TiN (low (500×) and high (5000×) magnifications, respectively). (**c**) and (**d**): CVD TiN (low (500×) and high (5000×) magnifications, respectively).

**Figure 5 materials-13-01145-f005:**
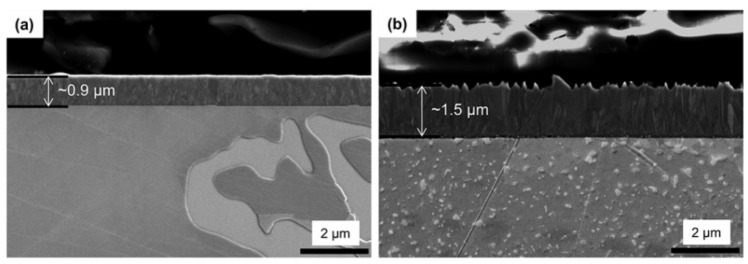
Cross-sectional SEM micrographs showing the PVD (**a**) and CVD (**b**) TiN coating layers on the mirror-polished Co–Cr alloy substrate (10000×).

**Figure 6 materials-13-01145-f006:**
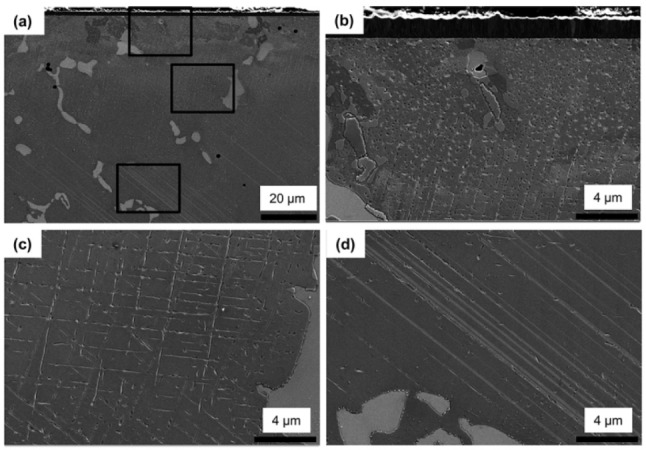
Low- (1000×, (**a**)) and higher- (5000×, (**b**–**d**)) magnification cross-sectional SEM micrographs (**a**) showing the CVD TiN coating layer and Co–Cr alloy substrate: approximately 12 μm (**b**), approximately 16 to 36 μm (**c**), and approximately 53 to 73 μm (**d**) from the coating interface.

**Figure 7 materials-13-01145-f007:**
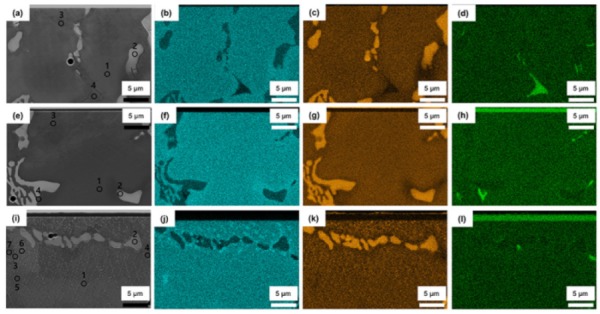
Cross-sectional SEM micrographs of the three groups (**a**,**e**,**i**) and their corresponding EDS mapping images ((**b**,**f**,**j**): Cr; (**c**,**g**,**k**): W; and (**d**,**h**,**i**): V) of the Co–Cr alloys (as-cast) (**a**–**d**) and PVD (**e**–**h**) and CVD (**i**–**l**) TiN-coated alloys (up to 20 μm from the interface, 3000×). The bright regions seen at the top of images (**h**,**l**) indicate Ti, which is superimposed with V. The circlets seen in images (**a**,**e**,**i**) indicate the points for the EDS spot analysis (see Table 2).

**Figure 8 materials-13-01145-f008:**
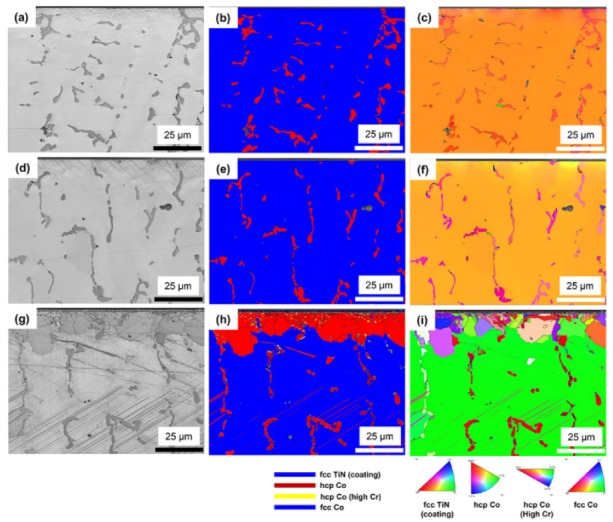
Cross-sectional electron backscattered diffraction (EBSD) images of the as-cast (**a**–**c**), PVD-coated (**d**–**f**), and CVD-coated (**g**–**i**) Co–Cr alloys: (**a**,**d**,**g**) band contrast (BC) maps; (**b**,**e**,**h**) phase maps; and (**c**,**f**,**i**) inverse pole figure (IPF) maps.

**Figure 9 materials-13-01145-f009:**
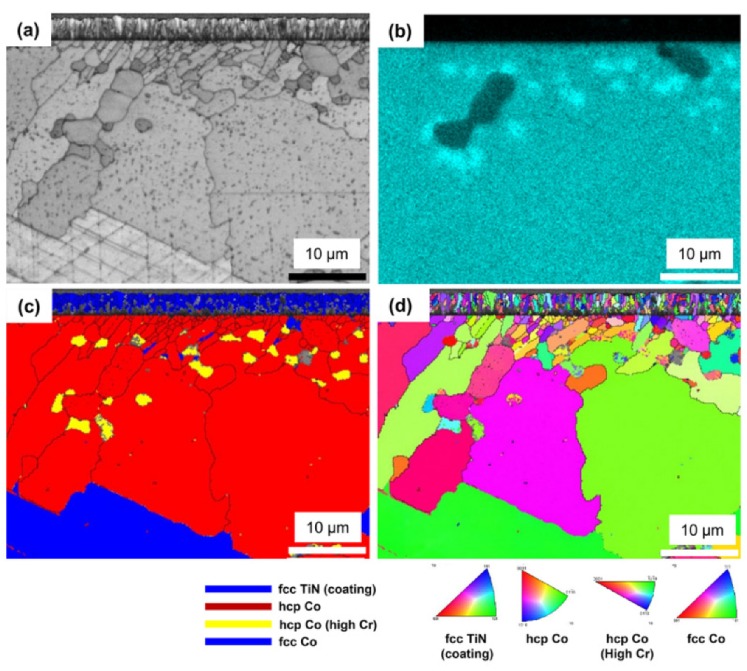
Cross-sectional high-magnification images of the CVD TiN-coated Co–Cr alloy analyzed using EBSD and EDS simultaneously: (**a**) BC map; (**b**) EDS map of Cr; (**c**) phase map; and (**d**) IPF map.

**Figure 10 materials-13-01145-f010:**
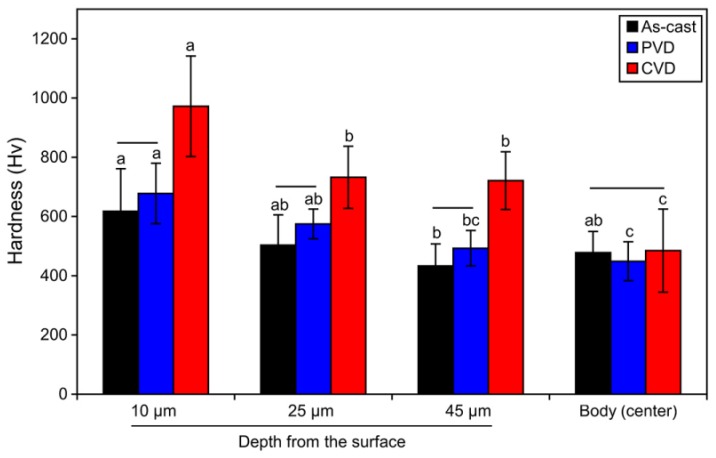
Nano-indentation hardness values measured at depths of 10, 25, and 45 μm and at the center of the cross-sectioned surface of the as-cast, PVD-coated, and CVD-coated alloys. Each value was converted into Vickers hardness (Hv) values. The horizontal bars connect statistically equivalent values among the three coating conditions within each depth condition (*p* > 0.05). The same small case letters above the bars indicate statistically similar means among the four depth conditions within each coating (*p* > 0.05).

**Figure 11 materials-13-01145-f011:**
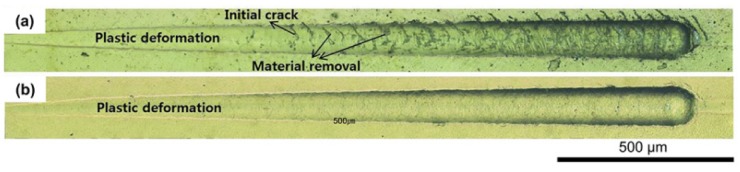
Progressive scratch test marks of the PVD-coated (**a**) and CVD-coated (**b**) Co–Cr alloy (maximum load of 30 N and length of 2 mm).

**Figure 12 materials-13-01145-f012:**
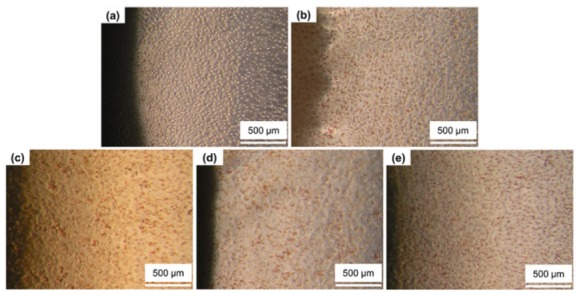
Morphology of L929 cells around the test specimens (100×): (**a**) positive control (rubber latex); (**b**) negative control (high-density polyethylene (HDPE)), (**c**) as-cast alloy, (**d**) PVD-coated alloy, and (**e**) CVD-coated alloy.

**Table 1 materials-13-01145-t001:** Main deposition conditions of physical vapor deposition (PVD) and chemical vapor deposition (CVD) coatings.

Coating	Temperature (°C)	Coating Time (min)	Pressure (mbar)	Bias Voltage (V)	Target	Gas
PVD	550	30	0.025	60	Ti	N_2_ 100%
CVD	850	400	150	N/A ^1^	N/A	H_2_ 59.5%; N_2_ 39.0%; TiCl_4_ 1.5%

^1^ Not applicable.

**Table 2 materials-13-01145-t002:** Results of the EDS spot analysis showing the atomic weight (at%) composition ratios for the points marked in images (a), (e), and (i) of Figure 7.

Image	EDS Composition (at%)							Remark
	Point	Co	Cr	W	Mo	Nb	V	
(a) As-cast	1	63.6	29.0	3.9	0.6	0.5	2.4	Co–Cr (matrix)
	2	49.6	20.5	12.9	2.6	12.7	1.7	W-Mo-Nb-rich
	3	64.6	28.5	3.7	0.6	0.4	2.2	Co–Cr (matrix white line)
	4	5.0	9.3	N/D ^1^	N/D	58.9	26.8	Nb-V-rich
(e) PVD-coated	1	62.9	29.4	3.9	0.7	0.7	2.4	Co–Cr (matrix)
	2	50.4	20.6	12.4	2.5	12.5	1.6	W-Mo-Nb-rich
	3	64.4	28.7	3.7	0.5	0.5	2.3	Co–Cr (matrix white line)
	4	7.2	9.7	0.8	N/D	56.4	25.8	Nb-V-rich
(i) CVD-coated	1	65.7	28.2	3.3	0.5	0.2	2.1	Co–Cr (matrix)
	2	51.5	19.4	11.1	2.8	13.7	1.5	W-Mo-Nb-rich
	3	64.8	29.3	2.9	0.5	0.3	2.2	Co–Cr (matrix small grain)
	4	8.4	10.4	0.3	N/D	56.1	24.9	Nb-V-rich
	5	48.3	43.2	3.5	1.2	0.8	3.0	Cr-rich
	6	59.9	27.2	7.5	1.4	1.6	2.6	White pit
	7	58.7	29.1	3.9	0.7	1.8	5.9	Black pit

^1^ N/D: not detected.

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
