# Peer review of "Application of a Novel CVD TiN Coating on a Biomedical Co–Cr Alloy: An Evaluation of Coating Layer and Substrate Characteristics"

_materials, 2020, doi:10.3390/ma13051145_

Round 1
Reviewer 1 Report
Review’s report for Materials for 728442
This manuscript describes preparation of Co-Cr alloy specimens coated with titanium nitride by chemical vapor deposition or physical vapor deposition, characterization of the specimens by X-ray analysis and SEM observation and evaluation of the hardness form the progressive scratch test. CVD-coated alloy showed a higher hardness value than PVD-coated ally, and no cracking and peering phenomenon after scratch test. Furthermore, the alloy showed no cytotoxicity. This approach for alloy preparation by CVD is very interesting. So, this manuscript is suitable for publication in “Materials”. However, the following points should be cleared before publication.
Major points;
- In Figure 11, CVD-coated alloy showed no cracking and peering phenomenon for scratch test with 30 N. I do not know whether 30 N is enough for orthopedic implant materials or not. The authors should describe the strength for biomaterial.
- In Figure 12, the cytotoxicity was evaluated from the morphology after treatment with the samples. However, I do not know the differences between the positive and negative controls. The authors should describe the reason for no cytotoxicity in detail.
Author Response
This manuscript describes preparation of Co-Cr alloy specimens coated with titanium nitride by chemical vapor deposition or physical vapor deposition, characterization of the specimens by X-ray analysis and SEM observation and evaluation of the hardness form the progressive scratch test. CVD-coated alloy showed a higher hardness value than PVD-coated ally, and no cracking and peering phenomenon after scratch test. Furthermore, the alloy showed no cytotoxicity. This approach for alloy preparation by CVD is very interesting. So, this manuscript is suitable for publication in “Materials”. However, the following points should be cleared before publication.
- We truly appreciate your encouraging comment.
In Figure 11, CVD-coated alloy showed no cracking and peering phenomenon for scratch test with 30 N. I do not know whether 30 N is enough for orthopedic implant materials or not. The authors should describe the strength for biomaterial.
- Since the hardness of the substrate, Co-Cr alloy, is much lower than that of the coating and the tip of the diamond cone used for the scratch test is sharp, the application of a high load may cause excessive deformation of the substrate, making it difficult to characterize the coating. In this study, the thickness of the CVD TiN coating was about 1.5 µm (Figure 5) and the deformation of at least 10 µm occurred in the depth direction when the scratch load was 30 N. This finding indicates that the loaded specimen underwent large strains, transferring large tensile and compressive stresses to the coating layer. When the maximum load was 10 N in our preliminary tests, both the PVD and CVD coating did not peel off. The test results for the maximum load of 50 N were similar to those for the 30 N load. Therefore, we concluded that 30 N was the optimum as the maximum load in our scratch test.
In Figure 12, the cytotoxicity was evaluated from the morphology after treatment with the samples. However, I do not know the differences between the positive and negative controls. The authors should describe the reason for no cytotoxicity in detail.
- Thank you for pointing it out. In our study, cytotoxicity of the alloys was examined using the agar-diffusion method. Thus, we examined the cell inhibition zones around the alloy specimens as well as the positive (rubber latex) and negative (HDPE) controls prior to the observation of the cell morphology. However, this aspect was not cleared addressed in our draft manuscript. We have added more detailed description on the inhibition zone to the text (Page 5, Lines 157–158; Page 13, Lines 367–369) in this revision.
Reviewer 2 Report
Comments and suggestions to the Authors:
The research is novel and important to the field. Abstract summarizes the research carried out in the manuscript, although it does not include the key findings, e.g. “The CVD coating showed superior adhesion to the PVD coating in the progressive scratch test” provide values or order of magnitude. The Introductory part is clear and well organized, the methodology is described sufficiently to reproduce the results. In opinion of this reviewer, the manuscript can be published in Materials after revision, addressing the following issues:
- I think providing SD in Table 2 will be helpful to confirm the statements such as “compositional analysis … revealed no significant difference in the amounts of matrix Co and Cr””
- Provide scale bars in Fig. 11 and Fig. 12.
The manuscript is somehow imbalanced: there is extensive characterization and discussion on the composition and mechanical properties and only 2(!) sentences describe the cytocompatibility tests. From the provided images it is impossible to assess the morphology of the cells, so the statement about “no morphological differences around the alloy” is not supported by the experiments.
- To assess the cytotoxicity, I suggest running at least MTT or XTT tests and provide figures with larger magnifications (preferably stained) to visualize the cells morphology.
The reference section is out-of-date, there are 27 refs among which 16 are older than 10 years from 90s and beginning of 00s. There are significant recent works on the topic, I strongly encourage the Authors to improve it.
OVERALL RECOMMENDATION: accept after minor revision
Author Response
The research is novel and important to the field. Abstract summarizes the research carried out in the manuscript, although it does not include the key findings, e.g. “The CVD coating showed superior adhesion to the PVD coating in the progressive scratch test” provide values or order of magnitude. The Introductory part is clear and well organized, the methodology is described sufficiently to reproduce the results. In opinion of this reviewer, the manuscript can be published in Materials after revision, addressing the following issues:
- Thank you very much for your encouraging comment and valuable suggestions.
I think providing SD in Table 2 will be helpful to confirm the statements such as “compositional analysis … revealed no significant difference in the amounts of matrix Co and Cr””
- Thank you for pointing it out. We did not repeat the EDS spot analysis several times enough to produce SD and statistical p-values. Therefore, we have slightly modified the sentence (Page 9, Lines 266–268).
Provide scale bars in Fig. 11 and Fig. 12.
- According to your suggestion, the scales bars have been added in the two revised figures.
The manuscript is somehow imbalanced: there is extensive characterization and discussion on the composition and mechanical properties and only 2(!) sentences describe the cytocompatibility tests. From the provided images it is impossible to assess the morphology of the cells, so the statement about “no morphological differences around the alloy” is not supported by the experiments.
- We agree with your comment. As stated in the Introduction section, the high coating temperature and some corrosive gas by-products generated during the CVD coating may cause an undesirable reaction at the implant-coating interface (Page 2, Lines 63–65). The present cytotoxicity test was performed as a preliminary evaluation of the potential cytotoxicity of the coating. We are planning to perform further experiments for confirming the cytocompatibility of the coating. The subsection (Pages 13–14, Lines 366–372) has been heavily modified according to your comment.
To assess the cytotoxicity, I suggest running at least MTT or XTT tests and provide figures with larger magnifications (preferably stained) to visualize the cells morphology.
- As stated above, a simple in vitro cytotoxicity test was performed in this study. According to your suggestion, we will perform further cytotoxicity evaluation of the coating using MTT or XTT tests, together with ICP analysis. In this revision, we have modified Figure 12, in which more enlarged photographs has been provided.
The reference section is out-of-date, there are 27 refs among which 16 are older than 10 years from 90s and beginning of 00s. There are significant recent works on the topic, I strongly encourage the Authors to improve it.
- We agree with your comment. To date, however, little research has been done on the application of CVD TiN coating to Co-Cr alloy (Page 2, Lines 68–69). Thus, we could not help depending on relatively old references. We have tried our best to update the reference list by replacing some of the old references with new ones (Page 15, Lines 413–423).